# Differential Participation of Plant Ribosomal Proteins from the Small Ribosomal Subunit in Protein Translation under Stress

**DOI:** 10.3390/biom13071160

**Published:** 2023-07-21

**Authors:** Zainab Fakih, Mélodie B. Plourde, Hugo Germain

**Affiliations:** Department of Chemistry, Biochemistry and Physics and Groupe de Recherche en Biologie Végétale, Université du Québec à Trois-Rivières, Trois-Rivières, QC G9A 5H9, Canada; zainab.fakih@uqtr.ca (Z.F.); melodie.bplourde@uqtr.ca (M.B.P.)

**Keywords:** *Nicotiana benthamiana*, *Arabidopsis thaliana*, translation regulation, ribosomal proteins from the small subunit (RPS), VIGS, 5′untranslated regions, transient expression, plant defense

## Abstract

Upon exposure to biotic and abiotic stress, plants have developed strategies to adapt to the challenges imposed by these unfavorable conditions. The energetically demanding translation process is one of the main elements regulated to reduce energy consumption and to selectively synthesize proteins involved in the establishment of an adequate response. Emerging data have shown that ribosomes remodel to adapt to stresses. In *Arabidopsis thaliana*, ribosomes consist of approximately eighty-one distinct ribosomal proteins (RPs), each of which is encoded by two to seven genes. Recent research has revealed that a mutation in a given single RP in plants can not only affect the functions of the RP itself but can also influence the properties of the ribosome, which could bring about changes in the translation to varying degrees. However, a pending question is whether some RPs enable ribosomes to preferentially translate specific mRNAs. To reveal the role of ribosomal proteins from the small subunit (RPS) in a specific translation, we developed a novel approach to visualize the effect of RPS silencing on the translation of a reporter mRNA (GFP) combined to the 5’UTR of different housekeeping and defense genes. The silencing of genes encoding for *NbRPSaA*, *NbRPS5A*, and *NbRPS24A* in *Nicotiana benthamiana* decreased the translation of defense genes. The *NbRACK1A*-silenced plant showed compromised translations of specific antioxidant enzymes. However, the translations of all tested genes were affected in *NbRPS27D*-silenced plants. These findings suggest that some RPS may be potentially involved in the control of protein translation.

## 1. Introduction

As sessile beings, plants have developed various strategies to overcome the range of challenging conditions they are exposed to. These responses are built on finely tuned gene expressions, which, in turn, lead to protein level variations. Changes in protein level depend on the regulation of multiple factors, such as transcription, mRNA structure, stability, transport, storage, protein synthesis, and degradation [1,2]. Among them, the translation process is one of the main elements that finely modulates protein accumulation under both biotic and abiotic stress situations; its regulation reduces energy consumption and allows for the selective synthesis of proteins involved in the proper establishment of an appropriate response [3,4]. Many examples of global translational inhibition and the preferential production of key proteins that are critical for adapting to environmental conditions are known [5,6,7,8]. A general decrease in global translation levels is observed in plants under conditions of sucrose starvation [9,10] and those acting in response to cold stress [11]. Furthermore, the overall translation activity in plants is higher in the light than in the dark; this is correlated with the higher energetic status of the plant cells under light conditions [12].

Protein synthesis is mediated by ribosomes and ribosomal-associated proteins. Ribosome assembly occurs within the nucleolus and requires the coordinated production and transport of four rRNAs (5S, 5.8S, 18S, and 28S) and eighty-one ribosomal proteins (RPs) [13]. The eukaryotic ribosome, termed the 80S ribosome, consists of two ribonucleoprotein subunits; the 40S small subunit binds the mRNA and provides the decoding site, which is formed by the 18S rRNA and thirty-three small ribosomal proteins (RPS). The 60S large subunit, which is composed of the 5S, 5.8S, and 23S rRNAs and 48 large ribosomal proteins (RPL), catalyzes the formation of peptide bonds [14,15,16]. All of these RPs are present in a single copy in each ribosome, except for the RPs forming a flexible lateral stalk on the large subunit [17,18]. In the model plant, *Arabidopsis thaliana*, each RP can be encoded by two to seven different members of the small families [15]. Thus, the 81 RP families may produce up to 10^34^ different potential ribosome structural conformations that could theoretically serve as a source of translation heterogeneity [3]. Although each RP gene has multiple paralogs, their expressions appear to be differentially regulated by various environmental cues and treatments with signaling molecules [19,20,21,22,23,24,25]. This differential expression between gene families, as well as within specific ribosomal gene families, opens vast possibilities for the functional role of these RPs in stress conditions. Furthermore, ribosome composition has, to date, been examined in several mass spectrometric studies, which have identified different r-protein paralogs within ribosomes that act in response to different stimuli, showing that ribosome composition may also be dynamic [3,26,27,28]. This heterogeneity can constitute specialized ribosomes that may regulate mRNA translation and control protein synthesis. Thus, specialized ribosomes are defined as a functional subpopulation of ribosomes that appear, for example, after an altered condition; they work to constrain translation to specific mRNAs and to shape the acclimated proteome [29].

The differential expression between the RP genes and the ribosomal composition implies a diversified functional relevance regarding RPs [30]. This is consistent with accumulating evidence that emphasizes the RP involvement in several ribosome functions, as well as roles away from the ribosome, such as DNA repair, histone binding, transcription-factors activity, and cell-cycle regulation [31,32,33]. For instance, mutations in some RPs influence the integrity of ribosomes, in structure and in function. The mutational analysis of several prokaryotic RPs has highlighted their importance in a variety of ribosomal processes. *RPS12* was shown to be required for tRNA decoding in the ribosomal A site [34] and the *RPS4* and *RPS5* mutations showed ribosome translational inaccuracy [35]; whereas, *RPSa*, *RPS7*, and *RPS11* are essential for mRNA binding [36]. Furthermore, in mammalian cells, the binding of the RACK1 (Receptor for Activated C-Kinase 1) to ribosomes is essential for the full translation of capped mRNAs and the efficient recruitment of eukaryotic initiation factor 4E (eIF4E) [37]. In *Nicotiana benthamiana* and *Arabidopsis thaliana*, QM/RPL10A plays a transcriptional role in regulating translational mechanisms and defense-associated genes [38]; also, *RPS27B* is involved in the degradation of damaged RNAs (induced by genotoxic treatments) [39].

The involvement and specific constitution of the protein-translation machinery in plant defense is poorly studied. Some reports have shown that the deficiency and mutation of ribosome proteins themselves are associated with disease responses in plants. The silencing of *RPL12* and *RPL19* in *N. benthamiana* and *A. thaliana* showed compromised nonhost disease resistance against multiple bacterial pathogens [40]; the silencing of *RPL10* in *N. benthamiana* and *A. thaliana* showed compromised disease resistance against the nonhost pathogen *Pseudomonas syringae* pv. *tomato* T1 [38]; and the silencing of *RPS6* in *N. benthamiana* affected the accumulation of the Cucumber mosaic virus, Turnip mosaic virus (TuMV), and Potato virus A (PVA), but not the Turnip crinkle virus and Tobacco mosaic virus [41].

Despite these studies, a systematic understanding of the functional role of RPs in the context of plant defense is still lacking. In the present study, using previously published nuclear proteomes of plants under stress, we identified several RPS that accumulated in the nuclei (the site of ribosome biogenesis) of stressed plants. We hypothesize that the accumulated RPS paralogs generate ribosomes that shape the cellular translatome and plant defense responses. To address the role of the identified RPS in a specific translation, we first developed a translation assay in which we tested the production of the green fluorescent protein (GFP) fused to different 5’UTR corresponding to known defense genes, or housekeeping genes, in the leaves of RPS-silenced and control plants. We found that three tested proteins (*RPSaA*, *RPS5A*, and *RPS24A*) are involved in the efficient translation of some defense proteins. In contrast, the protein *RPS27D* is involved in the general translational activity of the ribosome; whereas, *RACK1A* is involved in the efficient translation of several antioxidant enzymes. Our technical approach defines a suitable methodological strategy for testing ribosomal protein requirements for the translation of specific groups of mRNAs. Moreover, this suggests that RPS paralogs play a crucial role in translational control.

## 2. Materials and Methods

### 2.1. Plant Growth and Stress Treatments

Seeds of *N. benthamiana* were vernalized for 48 h at 4 °C and plants were grown in soil (AgroMix) at 23 °C and 60% relative humidity with a 14 h/10 h light/dark cycle in a growth chamber.

For 2,6-dichloroisonicotinic acid (INA) treatment, 3-week-old plants were sprayed to imminent runoff with an aqueous solution of 0.65 mM INA containing 0.05% Sylgard 309 surfactant; whereas, the mock treatment consisted of only the Sylgard 309 aqueous solution. Leaf tissues were harvested 24 h after being sprayed with INA, as previously described [42]. INA was used to induce plant defense as it was shown to induce a response similar to those of salicylic acid and pathogen infection [43]. For cold stress treatments, 3-week-old plants were placed at 4 °C for 6 h [44].

For the biotic stress experiments, we used the bacterial pathogen *P. fluorescens* EtHAn (Effector-to-Host Analyzer) strain, which allowed for the development of the PTI response in *N. benthamiana* [45]. The bacterial suspension of *P. fluorescens* EtHAn at OD_600_ = 0.2 in 10 mM of MgCl_2_ was infiltrated into the abaxial side of 3-week-old *N. benthamiana* leaves; tissue was collected 7 h post-inoculation. Leaf samples of infiltrated plants, with 10 mM of MgCl_2_ grown under similar conditions, were used as a control to normalize the expression. All of the samples were collected in the form of three biological replicates after each time interval and were immediately frozen in liquid nitrogen and stored at −70 °C.

### 2.2. Differential Gene Expression Analysis

A gene expression analysis in a *N. benthamiana* plant was performed on RNA extracted from the frozen tissue using the Genezol Total RNA kit (Geneaid), following the manufacturer’s instructions. The RNA quality was assessed by agarose gel electrophoresis and quantified by spectrophotometry. In total, 1 µg of each sample was used as the template for first-strand cDNA synthesis using the M-MuLV Reverse Transcriptase (New England Biolabs, Whitby, ON, Canada). Quantitative PCR amplification was performed on a CFX Connect detection system (Bio-Rad Laboratories, Mississauga, ON, Canada) using gene-specific primers and the SYBR Green PCR Master Mix (Bioline, Toronto, ON, Canada). The primers used were designed using the Primer 3 software; they were designed in such a way that they targeted a region that is completely absent of all other paralogous genes and is unique. This selection was performed using the VIGS Tool from the Sol Genomics Network (https://vigs.solgenomics.net/, accessed on 1 March 2018) (Appendix A). The specificity of the primers was then verified by using the Primer-Blast tool at NCBI. In total, a 100 ng cDNA template and 0.4 µM of each primer (listed in Appendix A) were used in a final volume of 20 µL. The amplification protocol included an initial denaturation at 95 °C for 2 min, with 40 cycles at 95 °C for 5 s, a primer-specific annealing temperature for 10 s, and an extension at 72 °C for 5 s. This was followed by constructing a melt curve at the end to estimate the amplification specificity of each gene. The data were analyzed with CFX Maestro qPCR software. *PP2A* and *UBQ1* (Polyubiquitin 1) were used as reference genes for normalization under INA conditions and those of a *P. fluorescens* EtHAn infection [46]. *ACT 2* and *UBQ1* were considered suitable genes to normalize with for the cold treatment [47]. The mean values of the relative fold change were calculated as per the ^ΔΔCt^ method [48]. RPS genes in each condition were defined as differentially expressed only if the expression value of the gene was more than 1.5-fold the control and had a *p*-value of less than 0.05 compared to the control.

The expression of the identified RPS genes in *Arabidopsis* was analyzed using the Genevestigator tool (https://genevestigator.com/, accessed on 6 February 2018) with the *Arabidopsis* Gene Chip platforms (ATH1: 22k array). The perturbation tool of the Genevestigator software was used to estimate the levels of gene expression as a heat map under different conditions. Data were presented as absolute log_2_ values of fold change compared with that of the control samples.

### 2.3. Virus-Induced Gene Silencing (VIGS)

The pBINTRA6 and pTV00 vectors were used for silencing in the *N. benthamiana*. The pTV00::*NbRPSaA*, pTV00::*NbRPS5A*, pTV00::*NbRPS27D*, pTV00::*NbRPS24A*, and pTV00::*NbRACK1A* constructs were developed and used for VIGS, as described [49]. In order to select VIGS silencing sections that were specific to a single paralog of the targeted protein, we used the VIGS Tool from the Sol Genomics Network (https://vigs.solgenomics.net/, accessed on 1 March 2018); we were able to design VIGS fragments unique to the 3′UTR of each targeted gene that was absent from the other paralogs (Appendix A). Appendix A provides a list of all of the paralogs of the investigated RPS. PCR was used to amplify the desired fragments with specific primers (Appendix A) using genomic DNA prepared from the plant tissues. The amplified fragments of the RPS genes and the pTV00 vectors were digested by the restriction enzymes *Kpn*I and *Hind*III, according to the manufacturer’s instructions; the purified products of the RPS sequence were inserted into the pTV00 vectors using T4 DNA ligase (NEB, England). The vectors were then transformed into competent cells of the *E. coli* strain DH5α. The selected positive clones with the correct sequence were used to transform the *Agrobacterium tumefaciens* strain of GV3101 electrocompetent cells. Plant infiltration was performed, as described previously [49]. The *Agrobacterium* strains of GV3101 containing pTV::*NbRPSaA*, pTV::*NbRPS5A*, pTV::*NbRPS27D*, pTV::*NbRPS24A*, or pTV::*NbRACK1A* and those of C58C1 containing pBINTRA6 were grown at 28 °C in a liquid Luria-Bertani medium including antibiotics (50 μg mL^−1^ kanamycin and 50 μg mL^−1^ rifampicin). After 24 h, the cells were harvested by centrifugation and resuspended in the infiltration buffer (10 mM of MgCl_2_ with 200 μM of acetosyringone and 10 mM of MES, pH 5.6) to a final optical density, at 600 nm, of approximately 0.5 and were agitated for 2 h (28 °C) before mixing in a 1:1 ratio. The *Agrobacterium* mix, containing either pBINTRA6 or pTV-*NbRPS* vectors, was infiltrated using a needleless 1-mL syringe that was inserted into the lower leaves of 2-week-old *N. benthamiana* plants [50]. As a control, the empty cloning vector pTV was used to distinguish the nonspecific phenotypic effects of VIGS.

### 2.4. Quantitative RT-PCR

Leaf tissue was collected 3 weeks after TRV inoculation to test the downregulation of ribosomal protein-encoding gene transcripts in *N. benthamiana*-silenced plants. The total RNA was extracted from silenced and mock-infiltrated plants and the first-strand cDNA was synthesized with oligo(dT_15_) primers using M-MuLV Reverse Transcriptase (New England Biolabs, Whitby, ON, Canada), according to the manufacturer’s instructions. The RT-qPCR was performed using the CFX Connect detection system (Bio-Rad Laboratories, Mississauga, ON, Canada). *ACT 1* and *EF1α* were used to normalize the transcript levels [51]. Each sample was run in triplicate and repeated six times from two pooled biological replicates of silenced and non-silenced plants. The average of the six experiments was calculated and the results were graphed, with the corresponding standard deviations indicated with bars in the figures. The primers used in this study are listed in Appendix A.

### 2.5. 5′UTR Chimeras and Plasmid Construction

The Cauliflower mosaic virus (CaMV) 35S promoter (p35S) and 5′UTR fusion constructs were assembled by PCR stitching. Briefly, two rounds of a PCR were carried out. In the first round, two separate PCRs were performed: one amplified the p35S from the *pB7FWG2* vector using specific primers listed in Appendix A*;* the other amplified the 5′ upstream region of 5 defense genes, or 3 housekeeping genes, from the *N. benthamiana* genomic DNA using gene-specific primers (Appendix A). The selection of these genes and the categorization of housekeeping and defense genes were made following a literature review. For instance, the *PP2A*, *F-BOX*, and GAPDH genes were consistently reported as housekeeping genes within the context of different viral infections in *Nicotiana benthamiana* [51,52,53] and under different conditions in other species [54,55,56,57]. Furthermore, the catalase, peroxidase, and ascorbate peroxidase proteins play a crucial role in overcoming various stress conditions and work as part of the antioxidant defense system [58]. In addition, the NPR1 (NONEXPRESSOR OF PR1) protein functions as a master regulator of plant hormone salicylic acid (SA)-signaling and plays an essential role in promoting defense responses [59]. Finally, the MAPK3 protein is implicated in stomatal development, biotic stress responses, and abiotic stress responses and is required for the complete “priming” of plants [60]. The mRNAs encoding these proteins showed a status indicating a higher translational efficiency in response to stress [61,62,63]. The 5′UTRs of these genes were identified using the Sol Genomics *N. benthamiana* draft genome (https://solgenomics.net/organism/Nicotiana_benthamiana/genome, accessed on 3 June 2023). In the second round, the products of these two PCRs, which overlapped at one end, were subsequently mixed and amplified.

Amplified fragments containing the p35S promoter and the 5′UTR were used to generate expression vectors, having different 5’UTRs linked to the reporter gene GFP. Amplicons were inserted into the pDONR221 vector (Invitrogen, part of Thermo Fisher Scientific, Waltham, MA, USA) via BP recombination reactions and then into the plant-expression vector PBGWFS7 via LR recombination reactions using Gateway technology [64].

### 2.6. Leaf-Infiltration Method

For transient GFP protein expression, constructs were introduced into the *A. tumefaciens* strain GV3101 by electroporation and were delivered into the leaf cells of silenced and non-silenced *N. benthamiana* (5-week-old) using the agroinfiltration method, as previously described [65]. Briefly, recombinant bacterial strains were grown overnight in a liquid Luria-Bertani medium with spectinomycin (50 mg/L); then, they were harvested and resuspended into an infiltration buffer (10 mM of MgCl_2_ and 150 μM of acetosyringone) to obtain a 0.5 unit of optical density at 600 nm. One hour after resuspension, leaves were infiltrated on their abaxial side. To minimize leaf-to-leaf variation, each leaf was infiltrated with a vector containing the 5′UTR of two housekeeping genes (*F-box* and *PP2A*) as normalization controls, alongside vectors containing the 5′UTRs to be tested. Three independent infiltrations were made for each experiment and were compared using the Student’s *t*-test. Ultimatley, *p* < 0.05 was represented with one star (∗). The agro-infected leaves were collected at 5 days post-infiltration to be photographed and analyzed for GFP production by spectrofluorimetry.

### 2.7. Detection of GFP Fluorescence

Leaves producing GFP were photographed under UV illumination generated by a 100 W, hand-held, long-wave UV lamp (Model B-100, UVP, Upland, CA, USA). The GFP fluorescence intensity was quantified at an excitation of 485 nm and an emission of 538 nm using a Synergy H1 Microplate Reader, BioTek, as described by Diamos et al. [66]. GFP samples were prepared by a serial two-fold dilution with phosphate-buffered saline (PBS, 137 mM of NaCl, 2.6 mM of KCl, 10 mM of Na_2_HPO_4_, and 1.8 mM of KH_2_PO_4_, pH 7.4); 100 μL of each sample was added to black-wall 96-well plates (Thermo Fisher Scientific), in triplicate. All measurements were performed at room temperature and the reading of an extract from an uninfiltrated plant leaf was subtracted before graphing. A standard curve of fluorescence for the GFP concentration was generated by measuring the fluorescence of a dilution series of GFP (triplicate) in a 96-well plate in the plate reader.

### 2.8. Protein Extraction

Total protein extract was obtained by homogenizing agroinfiltrated leaf samples with a 1:5 (*w*:*v*) ice-cold extraction buffer (25 mM of sodium phosphate, pH 7.4, 100 mM of NaCl, 1 mM of EDTA, 0.2% Triton X-100, 10 mg/mL of sodium ascorbate, 10 mg/mL of leupeptin, and 0.3 mg/mL of phenylmethylsulfonyl fluoride) using a mortar and pestle. To enhance solubility, homogenized tissue was rotated at room temperature for 30 min. The crude plant extracts were clarified by centrifugation at 10,000× *g* for 10 min at 4 °C.

## 3. Results

### 3.1. Small Ribosomal Proteins Are Deregulated by Different Stresses in A. thaliana and N. benthamiana

Ribosome biogenesis represents a compendium of steps by which the ribosomes may become assembled, involving the import of most RPs into the nucleus and nucleolus and their association with rRNA to constitute the ribosomal subunits [13]. Hence, many studies have identified ribosomal proteins in the nuclei of various plants under stress [44,67,68,69,70]. In line with this idea, we analyzed the published datasets on the biotic and abiotic stress-responsive nuclear proteomes in various plant species to identify plant ribosomal proteins of the small subunit involved in disease resistance and selected RPS detected in the nuclei of stressed plants [44,67,68]. A previous proteomics analysis identified a subset of 11 RPS detected in the nuclei of elicited immunity in *Arabidopsis* plants following a chitosan elicitor treatment [44,67,68] (Appendix A). In response to cold stress, eight RPS were overrepresented in the nuclear proteome of *Arabidopsis* [44] (Appendix A). Furthermore, seven RPS had a significant change in abundance in the nucleus of a tomato (*Solanum lycopersicum*) during an infection caused by the oomycete pathogen *Phytophthora capsici* [68] (Appendix A). From these three studies, a total of 15 different RPS displayed an increased nuclear abundance under various stress conditions.

The gene-expression patterns of these 15 RPS were evaluated under stress conditions in *Arabidopsis* using the Genevestigator application’s compendium of microarray experiments. We evaluated the expression of these genes following cold stress (Appendix A), elicitor treatment (Appendix A), and biotic stress (Appendix A). Genes that showed a strong induction in at least two conditions were selected as upregulated. Interestingly, six genes (*RPSaA*, *RPS10C*, *RPS12C*, *RPS19C*, *RPS27D*, and *RACK1A*) showed high levels of expression in response to all three stresses (Appendix A).

To gain insights into the expression patterns of RPS genes in *N. benthamiana* plants in stress contexts and, also, to provide a comparative analysis of the RPS expression between the two model plants, we performed a quantitative reverse transcription qRT-PCR of the 15 RPS genes using *N. benthamiana* tissues with cold stress conditions (Figure 1a), an INA treatment, an analog of SA that induces plant defense [43] (Figure 1b), and infection with the bacteria *Pseudomonas fluorescens EtHAn* (Figure 1c). Five genes (*RPSaA*, *RPS5A*, *RPS24A*, *RPS27D*, and *RACK1A*) were highly regulated under the three stress treatments (Figure 1d). It is worth mentioning that the expression patterns of *RPSaA*, *RACK1A*, and *RPS27D* in *N. benthamiana* are consistent with the *Arabidopsis* data from the Genevestigator microarray database. We herein focus on these five RPS genes for further functional analyses in *N. benthamiana*.

### 3.2. RPSaA, RPS5A, and RPS24A Proteins Are Involved in the Translation of Defense Proteins Encoding mRNAs

To test whether the silencing of a specific *NbRPS* gene compromises defense genes’ translations, direct measurements of chimeric reporter mRNA translational efficiencies were compared between RPS-silenced and mock-infiltrated plants. Each chimeric mRNA contained the 5’upstream region of either a defense gene or a housekeeping gene fused to the coding sequence of the green fluorescent protein (Table 1). All silenced plants showed more than a 50% down-regulation of the target transcripts (Figure 2a). Then, the different chimeric constructs were delivered into the *N. benthamiana* leaves of silenced and control plants by an *A. tumefaciens*-mediated transformation and the green fluorescence was monitored. To minimize leaf-to-leaf variation, each leaf was infiltrated with a vector containing the 5’UTR of two housekeeping genes, *F-box* and *PP2A,* as controls alongside vectors containing the 5’UTRs to be tested. No fluorescence was detected in the plant leaves infiltrated with empty vectors without any 5’UTRs (PBGWFS7 vector); whereas, a significant GFP fluorescence was observed with all of the 5’UTR-GFP chimeras in the control plants (Figure 2b). Using this system, we found that the GAPDH 5’UTR construct produced intense green fluorescence in both silenced and control plants; whereas, the constructs containing the 5’UTRs of catalase, peroxidase, ascorbate peroxidase, NPR1, and MAPK3 showed poor GFP signals in pTV::*NbRPSaA*, pTV::*NbRPS5A*, and pTV::*NbRPS24A* compared to the mock plant (Figure 2b). GFP fluorescence was quantified by spectrofluorimetry and was decreased by more than 50%; sometimes it was almost absent, particularly for the defense chimeric constructs in RPS-silenced plants (Figure 2c–e). These results indicated that *NbRPSaA*, *NbRPS5A*, and *NbRPS24A* are essential for the optimal translation of many defense genes in planta. 

### 3.3. RPS27D Is Required for Efficient Translation in N. benthamiana

The ribosomal protein *S27* (*RPS27*), belongs to the 40S subunit and, through its zinc-finger-like motif, it acts as an RNA-binding protein and subsequently influences the transcription of many genes through transcript degradation [39]. *A. thaliana* and *N. benthamiana* both have four *RPS27* gene family members: A, B, C, and D. The amino acid similarity between *AtRPS27* and *NbRPS27* proteins is between 89.8 and 96.5% (Figure 3a) [26]. The alignment of the *S27* ribosomal protein sequences of different species (rice, barley, rat, and human) shows high conservation [39]. We. sought to investigate the role of *NbRPS27D* in the translation of defense genes in *N. benthamiana* using the same TRV-mediated virus-induced gene-silencing approach to downregulate *NbRPS27D* expression. The silenced plants showed a more than 60% down-regulation of the target transcript compared to the control (Figure 3b). We then tested translational efficiency by analyzing the GFP accumulation from the agro-infiltration of chimeric mRNAs. *NbRPS27D* silencing resulted in a more than 50% decrease in GFP production in the zone of infiltration, with the vectors containing the 5′UTRs of defense genes (Figure 3c,d). Interestingly, a similar decrease was observed for the GFP vector containing the 5′UTR of the housekeeping gene, GAPDH (Figure 3c,d). The data presented here suggest that one paralog of ribosomal protein *S27* (*RPS27D*) may play a crucial role in the ribosome translational activity in *N. benthamiana*.

### 3.4. RACK1A Is Required for the Efficient Translation of Several Antioxidant Enzymes

RACK1 was originally isolated as a receptor for activated C-kinase 1. In addition to its signaling roles, it interacts with the ribosomal machinery, several cell surface receptors, and nuclear proteins [75]. The most stable and consistent interaction of RACK1 is the one it has with the ribosome. Indeed, RACK1 is found at the surface exposed region of the 40S ribosomal subunit, next to the mRNA exit channel [14,76,77]. It is known that RACK1 specifically modulates translational efficiency in various model systems [37,78,79]; however, its role in the efficient translation of mRNA subsets in the context of defense in planta is not well characterized. *N. benthamiana* has five RACK1 homologs [26] and *A. thaliana* has three [74]. *AtRACK1A* and *NbRACK1A* share 82% of their amino acid identities (Figure 4a). Since *RACK1A* is the paralog that was previously detected in the nucleus of stressed plants (Appendix A), we silenced *NbRACK1A* to investigate its role in the translation of defense genes. Additionally, qRT-PCR analyses confirmed a down-regulation of more than 70% of the targeted transcript in the silenced plants compared to the control (Figure 4b). *NbRACK1A* silencing caused an important decrease in the GFP fluorescence in leaves infiltrated with the vectors containing the 5′UTRs of peroxidase, ascorbate peroxidase, and catalase (Figure 4c). Similarly, the quantification data indicate that the GFP production under these 5′UTRs was decreased in the pTV::*NbRACK1A* plants compared to control plants (Figure 4d). By contrast, *NbRACK1A* silencing had no effect on GFP production in the areas infiltrated with the other 5′UTRs (Figure 4c,d). These results suggest that *NbRACK1A* silencing in *N. benthamiana* compromises the translation of several antioxidant enzymes.

## 4. Discussion and Conclusions

Plants’ responses to stress vary according to stress-type and the outcome is mainly specific to a particular stress [80,81]. Recently developed technologies, such as ribosome profiling and quantitative proteomics, have shown that many stresses inhibit protein synthesis in cells [2,5,6,82]. Protein synthesis accounts for a large proportion of the energy budget of a cell and, thus, requires tight regulation [83]. However, a severe reduction in translation can be harmful during stress as it is precisely the time when cells require new protein synthesis in order to repair damage and adapt to the new environment [84]. Thus, selective translation regulation may allow cells to react to adverse conditions more effectively. As translation modulation is a fast response to environmental signals, the ribosome could be a player in this adaptation. It has been shown that translational regulation mainly takes place during the initiation steps [85]. In plants, the initiation of a translation requires initiation factors, mRNAs, tRNAs, and ribosomes. It involves numerous protein–RNA and protein–protein interactions. Briefly, the 40S subunit of the ribosome directly binds to mRNAs in a way that is dependent on the mRNAs’ structures. After mRNA binding and scanning to the AUG start codon in a favorable context, the 60S subunit is recruited to form an 80S initiation complex capable of entry into the elongation phase [86]. During the initiation process, mRNA recruitment to the 40S ribosomal subunit is thought to be the rate-limiting step and is often modulated. The translation efficiency is determined by structural features in the 5′ untranslated region (5′UTR) of the mRNA. These features not only determine how well an mRNA is translated but also whether specific ribosomal proteins and other proteins can interact with it [87].

Recent studies on *Arabidopsis* ribosomes revealed that numerous r-proteins are represented by two or more gene family members and most members of each family are expressed [29,88]; meanwhile, r-proteins are generally found as a single copy per ribosome [17,89]. However, the expression of each RP gene appears to be differentially regulated by different conditions. In line with this idea, changes in the expression patterns of these 15 RPS genes in *Arabidopsis* and *Nicotiana* were compared in order to gain insights into the regulation of the response to stress. We have shown differential expression under the three stress treatments of these seven (*RPSaA*, *RPS10C*, *RPS12C*, *RPS19C*, *RPS21C*, *RPS27D*, and *RACK1A*), five (*RPSaA*, *RPS5A*, *RPS24A*, *RPS27D*, and *RACK1A*), and three (*RPSaA*, *RACK1A*, and *RPS27D*) RPS genes in *Arabidopsis*, *Nicotiana*, and both plants, respectively. Similarly, several r-protein genes have been found to be upregulated under different stimuli in plants [19,20,21].These results indicate the fundamental stress-specific reprogramming of RP gene transcription under stress conditions. 

With that in mind, it is interesting to speculate that these RPS endow ribosomes with the capacity for preferential mRNA selection for translation. To substantiate this hypothesis, using the VIGS method, we prepared plants with reduced levels of *RPSaA*, *RPS5A*, *RPS24A*, *RPS27D*, and *RACK1A* mRNAs and tested the translation efficiency of specific groups of selected mRNAs. *NbRPSaA*-, *NbRPS5A*-, and *NbRPS24A*-silenced *N. benthamiana* plants showed varying extents of compromised translation when compared to control plants. The green fluorescence intensity from the GAPDH 5’UTR-GFP chimera was similar in *RPSaA*-, *RPS5A*-, and *RPS24A*-silenced leaves and non-silenced control plants. However, the silencing of these three RPS genes resulted in a dramatic reduction in the translations of catalase, peroxidase, ascorbate peroxidase, NPR1, and MAPK3 5’UTR-GFP chimeras. Interestingly, previous studies have reported that *RPSaA*, *RPS5A*, and *RPS24A* have roles in the regulation of reactive oxygen species (ROS)-mediated systemic signaling. *RPS5A* may play an important role in dark treatment by participating in the autophagy regulatory process, which is triggered to degrade excessive ROS to help protect cells [90]. Then, it was shown that *RPSaA* and *RPS24A* had lower degradation rates and were more stable after oxidative stress [91]. Thus, the transcript levels of *RPSaA*, *RPS5A*, and *RPS24A* increased significantly in vanilla infected with *Fusarium oxysporum* f. sp. *vanilla* [20]; additionally, the expression of *RPS5A* was also induced by *Xanthomonas oryzae* pv. *oryzae* and *Rhizoctonia solani*, rice pathogens that, respectively, cause very serious bacterial leaf blight and sheath blight diseases [19]. Moreover, the expression of *RPSaA*, *RPS5A*, and *RPS24A* was upregulated under the three stress treatments in *Nicotiana* in our study. A differential accumulation of *RPSaA*, *RPS5A*, and *RPS24A* in the ribosomal apparatus was reported in *Arabidopsis* following treatment with the defense-inducing compound INA [28]. The data obtained in this study demonstrate that *RPSaA*, *RPS5A*, and *RPS24A* are required for the selective translation of defense genes to cope with unfavorable conditions. 

Meanwhile, the data presented here suggest that *NbRPS27D* silencing leads to a general translation defect. As mentioned above, *RPS27* was shown to contain a conserved zinc finger motif, which may confirm its ability to interact with non-ribosomal components, especially mRNAs. In *Arabidopsis*, *RPS27B* has been said to act as a regulator of transcript stability in response to genotoxic treatments via the degradation of damaged RNAs [39]. Recent data have shown that mutations in *RPS27B* influence the integrity of ribosomes in both structure and function [90].

RACK1 is a highly conserved scaffold protein located at the surface exposed region of the 40S subunit, near the mRNA exit channel [92]. The recent identification of RACK1 as a core component of the small subunit of the ribosome suggests it possesses signaling functions, allowing for translation regulation in response to cell stimuli [93,94]. RACK1 regulates several signaling pathways by acting as a receptor for signaling proteins, such as the protein kinase C (PKC) family [95,96], and by controlling mRNA-specific translation [37,97]. Previous studies have reported on the role of *RACK1A* in plant immune signaling and hormone responses [98,99,100]. *RACK1A* has also been demonstrated to be a key regulator of reactive oxygen species (ROS)-mediated systemic signaling [99,101]. ROS are important and common messengers produced in response to various environmental stresses and are known to activate many MAPKs [102]. They are recognized as threshold-level signaling molecules that regulate adaptations to various biotic and abiotic stresses, i.e., the ROS level determines whether they will be defensive or destructive molecules, which is maintained by a balance between ROS-producing and ROS-scavenging pathways for normal cellular homeostasis [103]. Interestingly, RACK1 affects ROS levels and ROS levels also affect RACK1 gene expression [104]. It has been reported that the knockdown of endogenous RACK1 increases the intracellular ROS level following H_2_O_2_ stimulation in human hepatocellular carcinoma cells, leading to cell death promotion [105]. At the same time, Saelee et al. showed that *Penaeus monodon*-RACK1 protected shrimp cells from oxidative damage induced by H_2_O_2_ [106]. Núñez et al. have shown that RACK1 positively regulates the synthesis of cytoplasmic catalase, a detoxification enzyme induced by hydrogen peroxide treatment, and controls the cellular defense against the oxidative stress in the fission yeast *Schizosaccharomyces pombe* [97]. In contrast, rice RACK1 (*OsRACK1A*) has been shown to be involved in the immune response against pathogen attacks through enhanced reactive oxygen species (ROS) [99]. Our study is in line with *RACK1A* regulating the ROS in plants. The observation that *RACK1A* knockdown reduces the translation of catalase, APOX, and POX mRNAs is in agreement with RACK1’s positive regulation of the detoxification enzyme synthesis induced by ROS. These enzymes are part of the antioxidant machinery, which helps to mitigate oxidative stress-induced damage. With respect to the role of RACK1 in signaling, accumulating evidence suggests that free RACK1 can act as a signaling molecule at a threshold level to enhance the production of ROS. Overall, it is noteworthy that the signal activated by free RACK1 is transient because, in the absence of ribosomal binding, the protein is unstable [37]. In contrast, RACK1 as a ribosomal protein controls the cellular defense against oxidative stress, positively regulating the translation of specific gene products involved in detoxification [97]. In conclusion, we propose that the association of *RACK1A* with the ribosome may indeed be regulated downstream of the ROS burst in order to modulate its translation functions; however, this possibility should be further investigated.

Our findings would argue that ribosomal proteins function in a modular fashion to decode genetic information in a context-dependent manner. The silencing of one r-protein, as was conducted in this study, could impact the stability and efficiency of the entire ribosome. In our case, the controls showed us that the overall translation efficiency was not impacted and, therefore, we believe that some variable ribosomal proteins additionally function in a coordinated manner to shape the translatome, which is adapted to different environmental cues in plants.

Overall, our study clearly demonstrates that some RPS are involved in the optimal translation regulation of many genes that are important for defense. However, our experimental design does not allow us to rule out whether the results presented herein are paralog-specific or if the effects could be true for all paralogs (or more than one paralog) of the same ribosomal protein. The findings of this study provide a novel strategy to assess translation efficiency that opens new and interesting avenues for research about the roles of ribosomal proteins during biotic and abiotic stress in *N. benthamiana*. Based on these data, we anticipate that some of the previously described biological functions of these RPS in plant immunity might be linked to their function as putative translational regulators. Future studies into the connection between the RP-mediated translation of defense proteins and the broader role of paralog specificity may provide a novel perspective on specialized ribosomes and translational control in plant disease.

## Figures and Tables

**Figure 1 biomolecules-13-01160-f001:**
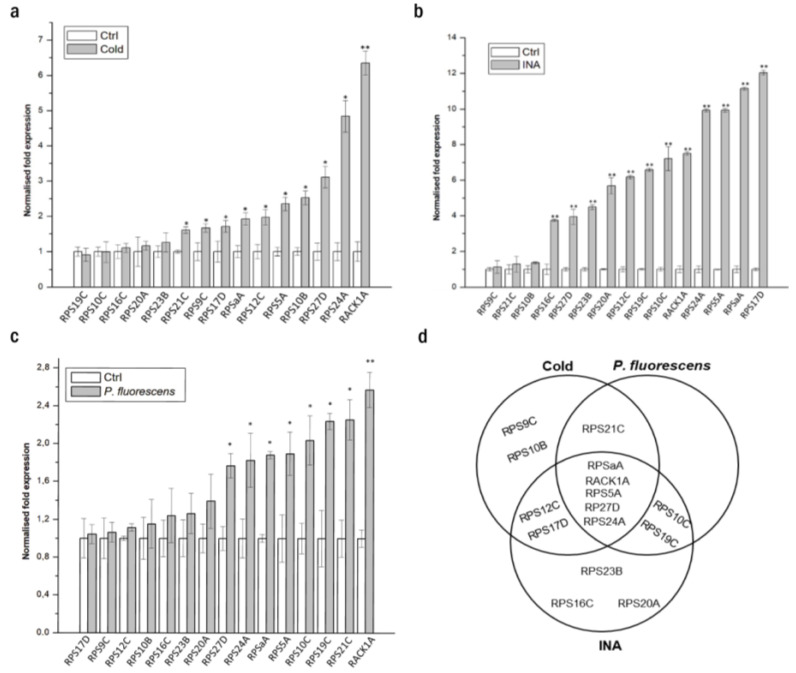
The mRNA levels of the ribosomal proteins of the small subunit are deregulated by different stresses in *N. benthamiana.* Relative expression of selected RPS genes in *N. benthamiana* following (**a**) cold stress, (**b**) INA treatment, and (**c**) *Pseudomonas fluorescens EtHAn* infection. * *p-*values < 0.05 and ** *p*-values < 0.01, Student’s *t*-test. (**d**) Venn diagram of the deregulated RPS genes under the three conditions.

**Figure 2 biomolecules-13-01160-f002:**
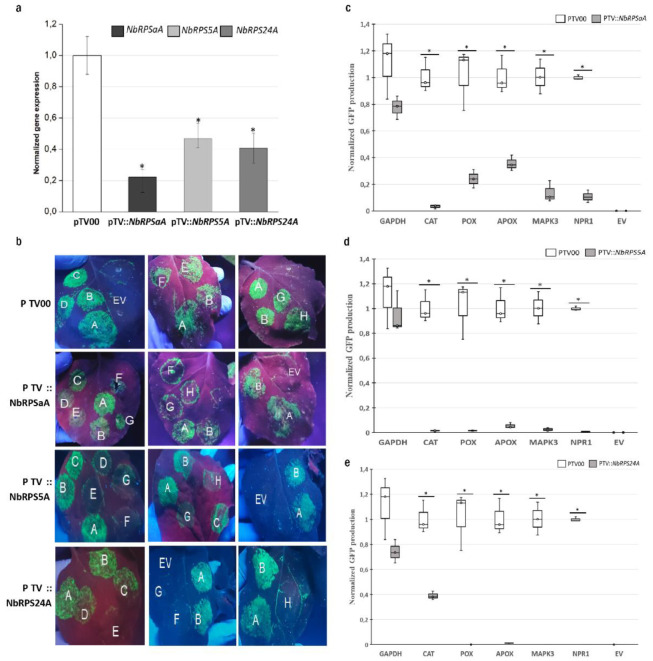
*RPSaA*, *RPS5A*, and *RPS24A* each have a role in the translation of defense proteins encoding mRNAs. (**a**) Relative expression levels of *NbRPSaA*, *NbRPS5A*, and *NbRPS24A* using quantitative RT-PCR analysis in the VIGS-treated *N. benthamiana* plants 21 days after agroinfiltration with TRV vectors. *ACTIN 1* and *EF1α* were used as internal references. Error bars represent the standard deviations of six independently infiltrated leaves from two biological replicates; asterisks (∗) indicate significant differences based on the Student’s *t*-test (*p* < 0.05). (**b**) GFP fluorescence in *N. benthamiana* leaves under UV light 5 days after infiltration (dpi), with *A. tumefaciens* carrying the p35S-5′UTR-GFP-expression cassettes (A: F-BOX, B: PP2A, C: GAPDH, D: CAT, E: POX, F: APX, G: NPR1, H: MAPK3, EV: Empty vector (PBGWFS7)). All leaves were infiltrated with the 5′ UTRs of two housekeeping gene vectors (A and B), in addition to the other vectors, as an internal control for leaf and plant variability. (**c**–**e**) Fluorimetric analysis of GFP accumulation. GFP fluorescence was quantified on ground tissue from three independently infiltrated leaves using a plate reader. Box plots show the replicate distributions in GFP concentration for each 5′UTR construct. The asterisks (∗) represent significant differences between silenced and mock-infiltrated samples, based on the Student’s *t*-test (*p* < 0.05).

**Figure 3 biomolecules-13-01160-f003:**
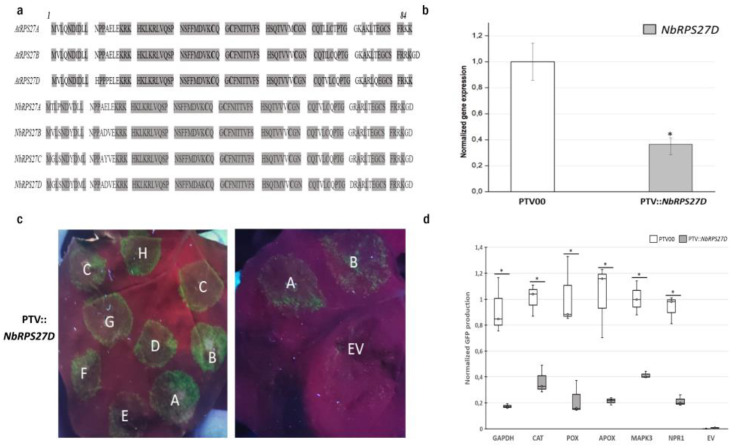
*RPS27D* is required for efficient translation in *N. benthamiana*. (**a**) Sequence alignment of *A. thaliana* and *N. benthamiana* ribosomal protein *S27* family members. Conserved amino acids in all the homologs are highlighted. Conserved cysteines forming a zinc finger are shown in bold. (**b**) Relative expression levels of *NbRPS27D* using quantitative RT-PCR analysis in the VIGS-treated *N. benthamiana* plants. *ACTIN1* and *EF1α* were used as references. Error bars represent the standard deviations of six independently infiltrated leaves from two biological replicates; the asterisk (∗) indicates a significant difference based on the Student’s *t*-test (*p* < 0.05). (**c**) GFP fluorescence in the *N. benthamiana* leaves of pTV::*NbRPS27D* plants under UV light 5 days after infiltration, with *A. tumefaciens* carrying the p35S-5′UTR-GFP-expression cassettes (A: F-BOX, B: PP2A, C: GAPDH, D: CAT, E: POX, F: APX, G: NPR1, H: MAPK3, EV: Empty vector (PBGWFS7)). All leaves were infiltrated with the 5′ UTRs of two housekeeping gene vectors (A and B), in addition to the other vectors, as an internal control for leaf and plant variability. (**d**) Fluorimetric analysis of GFP accumulation. GFP fluorescence was quantified on ground tissue from three independently infiltrated leaves using a plate reader. Box plots show the replicate distributions in GFP concentration for each 5′UTR construct. The asterisks (∗) represent significant differences between silenced and mock-infiltrated samples based on the Student’s *t*-test (*p* < 0.05).

**Figure 4 biomolecules-13-01160-f004:**
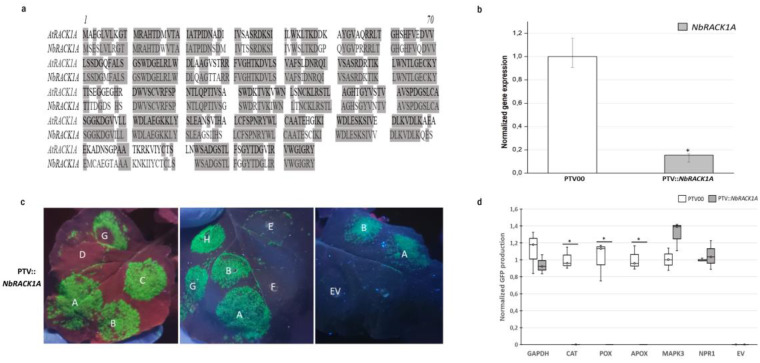
*RACK1A* is required for the efficient translation of several antioxidant enzymes. (**a**) Sequence alignment of *AtRACK1A* and *NbRACK1A*. (**b**) Relative expression levels of *NbRACK1A* using quantitative RT-PCR analysis in the VIGS-treated *N. benthamiana* plants. *ACTIN1* and *EF1α* were used as references. Error bars represent the standard deviations of six independently infiltrated leaves from two biological replicates; the asterisk (∗) indicates a significant difference based on the Student’s *t*-test (*p* < 0.05). (**c**) GFP fluorescence in the *N. benthamiana* leaves of pTV::*NbRACK1A* plants under UV light 5 days after infiltration, with *A. tumefaciens* carrying the p35S-5′UTR-GFP-expression cassettes (A: F-BOX, B: PP2A, C: GAPDH, D: CAT, E: POX, F: APX, G: NPR1, H: MAPK3, EV: Empty vector (PBGWFS7)). All leaves were infiltrated with the 5′ UTRs of two housekeeping gene vectors (A and B), in addition to the other vectors, as an internal control for leaf and plant variability. (**d**) Fluorimetric analysis of GFP accumulation. GFP fluorescence was quantified on ground tissue from three independently infiltrated leaves using a plate reader. Box plots show the replicate distributions in GFP concentration for each 5′UTR construct. The asterisks (∗) represent significant differences between silenced and mock-infiltrated samples based on the Student’s *t*-test (*p* < 0.05).

**Table 1 biomolecules-13-01160-t001:** List of the 5′UTRs used in this study.

Gene Symbol	Gene Name	Description	Reference
F-BOX	F-box protein	Normalizing gene	[52]
PP2A	Protein phosphatase 2A	Normalizing gene	[52]
GAPDH	Glyceraldehyde 3-phosphate dehydrogenase	Housekeeping gene	[52]
CAT	Catalase	ROS-scavenging enzymes	[71]
POX	Peroxidase	ROS-scavenging enzymes	[72]
APX	Ascorbate peroxidase	ROS-scavenging enzymes	[72]
MAPK3	Mitogen-activated protein kinases 3	PAMP-triggered immunity (PTI)	[73]
NPR1	Nonexpressor of Pathogenesis-Related Genes1	Positive regulator of SAR	[74]

## Data Availability

Data is contained within the article or Appendix A.

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
