# Peer review of "Differential Participation of Plant Ribosomal Proteins from the Small Ribosomal Subunit in Protein Translation under Stress"

_biomolecules, 2023, doi:10.3390/biom13071160_

Round 1

Reviewer 1 Report

This article provides a further example of the complexity of the ribosome and the potential role of its heterogeneity in the regulation of translation, and more specifically here, during stress in plants. It is well presented and the results seem to be supported by reliable experiments.
However, I feel it is necessary to adopt the universal nomenclature for naming ribosomal proteins (DOI: 10.1016/j.sbi.2014.01.002), so that all readers can fully grasp and understand the results. It would also be interesting to make a figure that shows specifically where proteins that exhibit particular behavior during different stresses are located in the 3D structure of the eukaryotic ribosome (DOI: 10.1126/science.1212642). Finally, I feel it is important to consider these results in the light of a recent concept concerning ribosomal proteins: they function collectively, forming a network that has become more complex over the course of evolution (doi: 10.1038/s41598-020-80194-4 ; doi: 10.3390/ijms20122911). Thus, local changes in r-protein composition can have a global effect on the connectivity and functionality of the whole

Author Response

Query 1: I feel it is necessary to adopt the universal nomenclature for naming ribosomal proteins (DOI: 10.1016/j.sbi.2014.01.002), so that all readers can fully grasp and understand the results.

Reply 1:  For the nomenclature of the ribosomal proteins, we have used the published annotations used by most studies on plant ribosome: Barakat et al., 2001; Carroll et al., 2008; Carroll et al., 2013; Hummel et al., 2015; Solano de la Cruz 2019; Eskelin et al., 2020; and we refer to TAIR database https://www.arabidopsis.org/browse/genefamily/athr.jsp. Although we are aware of the proposed nomenclature by Yusupov, it is our understanding that it has not yet been adopted by the plant community and we believe our reader will be mostly of the plant community. Because it is a special issue on ribosomes, and there may be a concern relating to uniformity we could agree to changing this nomenclature but we feel that it would be detrimental to our target audience. We specifically pointed our your comment to the editor since we believe that it is an editorial decision, we will abide by the decision of the editor.

Query 2:  It would also be interesting to make a figure that shows specifically where proteins that exhibit particular behavior during different stresses are located in the 3D structure of the eukaryotic ribosome (DOI: 10.1126/science.1212642).

Reply 2:  The figure of the localisation of the RPS in the ribosome of the model plant Arabidopsis is in the supplementary Figure (S2e) which the reviewer could not see because the Figure S2 was not in the final assembled PDF. 

Query 3:  I feel it is important to consider these results in the light of a recent concept concerning ribosomal proteins: they function collectively, forming a network that has become more complex over the course of evolution. Thus, local changes in r-protein composition can have a global effect on the connectivity and functionality of the whole.

Reply 3: This is a fair point which we addressed in the conclusion.

Reviewer 2 Report

General Overview:

The manuscript by Fakih and colleagues reports on an experimental tool to visualize the translational efficiency using virus-induced gene silencing (VIGS) coupled with transient expression of a reporter system (5’UTR-GFP) in N.benthamiana. The authors found a subset of small ribosomal proteins (RPSs) overrepresented in the abiotic and biotic stress condition based on the previous proteome research. The authors further confirmed mRNA expression levels of this subset in Arabidopsis and N.benthamiana under similar stress conditions using Genevestigator data and qPCR, respectively. The authors selected 5 overlapped genes (NbRPSaA, NbRPS5A, NbRPS24A, NbRACK1A, and NbRPS27D) and investigated the reporter translational efficiency in their silenced plants. The results from these assays indicate the differential engagement of RPS proteins depending on the 5’UTR sequences of mRNA. However, the major concern with the manuscript is that the authors only tested 8 genes, including controls. Thus, it is not possible to draw any conclusions regarding the common features of 5’UTR sequences regulated by specific RPS proteins.

Major points to be addressed to improve the manuscript.

1.     I was unable to find a pdf file that contains supplemental figures. Please provide the relevant supplemental figures for verification of the results.

2.     The authors should provide more relevant reasons for choosing the 5’UTR-constructed target genes. Is there any report indicating that these target genes are regulated at the translation level? If not, the authors should provide the translation efficiency of these genes under stress conditions using the reporter system.

3.     If NbRPSaA, NbRPS5A, NbRPS24A, and NbRPS27D are specifically involved in the efficient translation of antioxidant enzymes, the authors need to measure ROS levels using simple histochemical methods such as DAB staining and/or NBT staining.

4.     Similarly, if NbRPSaA, NbRPS5A, and NbRPS24A are specifically involved in the efficient translation of defense genes, the authors need to check defense responses such as pMAPK activity by PTI elicitors.

Minor points to be considered to improve the manuscript.

1.     Line 80: RACK1, need full gene name from line 378

2.     Line 286: Please include a sentence explaining the reason for INA treatment, moving it from the M&M section to here.

3.     Figures 2C-E, 3D, and 4D: catalase  CAT

4.     Figures 3A and 4A: Please use the same alignment tool for these figures.

Author Response

Query 1: I was unable to find a pdf file that contains supplemental figures. Please provide the relevant supplemental figures for verification of the result.

Reply 1:  The reviewer was right. The supplementary Figure S1 and S2 were missing in the assembled  PDF. They are included in the revision.

Query 2: The authors should provide more relevant reasons for choosing the 5’UTR-constructed target genes. Is there any report indicating that these target genes are regulated at the translation level? If not, the authors should provide the translation efficiency of these genes under stress conditions using the reporter system.

Reply 2:  We added the missing information about the translation efficiency under stress with 3 references in the text in the part 2.5 of materials and method.

Query 3: if NbRPSaA, NbRPS5A, NbRPS24A, and NbRPS27D are specifically involved in the efficient translation of antioxidant enzymes, the authors need to measure ROS levels using simple histochemical methods such as DAB staining and/or NBT staining.

Reply 3: We politely disagree with the reviewer. It has been demonstrated by others that catalase complete knockout results in undetectable changes in ROS levels notably hydrogen peroxyde; it is suspected that these mutations can be compensated by the many ROS-scavenging pathways present in plants (for references see Mhamdi 2010, Noctor 2002, Chaouch, 2010). Thus performing the experiment is likely to lead to a negative result.

Query 4: Similarly, if NbRPSaA, NbRPS5A, and NbRPS24A are specifically involved in the efficient translation of defense genes, the authors need to check defense responses such as pMAPK activity by PTI elicitors.

Reply 4: The reviewer brings a valid point, however there are two reasons why we believe we should not perform this experiment: 1) Similar to ROS scavenging, PTI is a highly robust and redundant pathway and measuring the output of PTI, on callose deposition for instance, is unlikely to result in significant changes.  Specifically measuring MAPK phosphorylation as suggested would require a minimum of 2-3 months (growing the plants, performing the VIGS, getting and troubleshooting the pAntibodies, etc.), it is impossible to perform this experiment in the 10 days allowed for resubmission.

Minor points to be considered to improve the manuscript.

Query 1: Line 80: RACK1, need full gene name from line 378

Reply 1:  This has been added.

Query 2: Line 286: Please include a sentence explaining the reason for INA treatment, moving it from the M&M section to here.

Reply 2:  This has been modified.

Query 3:  Figures 2C-E, 3D, and 4D: catalase à CAT

Reply 3:  All occurrences were modified as requested.

Query 4: Figures 3A and 4A: Please use the same alignment tool for these figures

Reply 4:  All occurrences were modified as requested.

Round 2

Reviewer 2 Report

I agree that if MDPI only provides 10 days for resubmission, those suggested experiments will not be able to perform within that period. However, to improve the manuscript in the following studies, I still believe that Q3 and Q4 will be necessary. I understand the authors’ perspective regarding the compensatory mechanisms in plants for ROS-scavenging pathways. However, I still believe that measuring ROS levels would provide valuable insights into the specific involvement of NbRPSaA, NbRPS5A, NbRPS24A, and NbRPS27D in the efficient translation of antioxidant enzymes.

The authors assume that performing the suggested experiments will likely lead to negative results because no remarkably increased levels of H2O2 were observed in cat2 mutants exposed to moderate irradiances. However, according to the suggested studies, increased H2O2 levels were detectable under stress conditions such as high-light stress or infection. Since the authors found that these RPS genes were up-regulated in the stress conditions, it is still worthwhile to check for increased ROS levels of VIGS plants in these stress conditions, even if ROS levels may not be clearly detectable in the normal conditions.

Furthermore, it is important to highlight that NbRPSaA, NbRPS5A, NbRPS24A, and NbRPS27D VIGS plants showed a significant decrease in the translation level of a "subset" of antioxidant enzymes, not just a catalase enzyme. Therefore, it is crucial to further explore the involvement of these RPS genes in the translation of antioxidant enzymes.

Author Response

We appreciated the constructive comment made by the reviewer, the proposed experiment is relevant but it is not possible to conduct it in 10 days.